# Current Concepts of Local Antibiotic Delivery in Bone and Joint Infections—A Narrative Review of Techniques and Clinical Experiences

**DOI:** 10.3390/microorganisms13102276

**Published:** 2025-09-29

**Authors:** Christof Ernst Berberich

**Affiliations:** Department of Medical Affairs, Heraeus Medical GmbH, 61273 Wehrheim, Germany; christof.berberich@heraeus.com

**Keywords:** prosthesis-related infections, infectious bone diseases, local anti-infective agents, prophylaxis, treatment, biocompatible materials, bone cements, bone substitutes, calcium sulfate, calcium phosphate, bioglass, hydrogels, intra-articular, intraosseous

## Abstract

Prophylactic measures and treatment strategies of implant-related bone and joint infections frequently involve the local delivery of high doses of antimicrobial drugs into the affected bone tissue or articular space in addition to the use of systemic antibiotics. Antibiotic-loaded biomaterials, such as Polymethyl Methacrylate (PMMA) cement, calcium sulfate, calcium phosphate, bioglass, and others, have proven to be clinically effective. However, they suffer from important limitations regarding elution and freedom of choice of admixable antimicrobial drugs. In order to overcome these shortcomings, the techniques of direct intraosseous or intra-articular injection/infusion of antibiotics via needles/cannulas or catheters are gaining popularity. Their attractiveness is based on the potential to achieve extremely high drug concentrations in situ, which can be maintained for as long as the catheters are left in place without increased risks of systemic toxicity. Although these methods are still in an experimental stage, reports on their clinical outcomes look promising. This articles summarizes the knowledge of when, how, and in which clinical settings the different modes and philosophies of local antibiotic delivery work best, with the aim to provide surgeons and infectious disease physicians guidance in clinical practice. This will help to optimize the use for the sake of the patients.

## 1. Introduction

The prevalence of musculoskeletal disorders in ageing societies is rising, leading to a growing need for surgical treatment [1]. Infections are major complications of traumatic bone injuries and orthopedic surgeries, causing severe morbidity among patients. Cures for the most serious bone and joint infection pathologies, such as osteomyelitis (OM), fracture-related infections (FRI), or prosthetic joint infections (PJI), are often hampered by the presence of antibiotic-resistant bacteria encased in the protective extracellular matrix of biofilms around implants and at the site of necrotic bone tissue [2,3]. Indeed, the chronic nature of biofilm infection with antibiotic-refractory persisters and intracellular bacteria makes the treatment of such infections very demanding; ideally, it requires a multidisciplinary approach. The expertise from various medical specializations, including surgeons, microbiologists, infectious disease specialists, pathologists, and others, is therefore often necessary [4,5].

Skilled surgery with radical debridement of the infected tissues crucial to reduce the initial bacterial load and microbial biofilms. Antibiotics then enhance the eradication of the germs, and functional reconstruction can be planned as part of a second step, if needed. A targeted antimicrobial therapy requires knowledge of the bacterial pathogens, agreement on the surgical strategy (e.g., cure vs. suppression), and consideration of patient risk factors [6,7,8]. However, the majority of the clinical studies and guidelines for antibiotic treatment of bone and joint infections have rather focused on systemic drugs, and evidence is often not of high quality. This leads to the observation that practice patterns are not standardized for a given clinical setting in the medical community and vary considerably among hospitals and countries [9,10].

There is scientific support for the assumption that antibiotics given at higher doses are related to better infection control than those given at lower doses by optimizing the pharmacokinetic and pharmacodynamic (PK/PD) parameters and by lowering the risk of bacterial resistance emergence [11]. Initial intravenous (IV) administration with loading doses followed by continued infusion has the advantage of more quickly achieving the target dose and bypassing the intestinal and hepatic first-pass effect or malabsorption problems with oral drugs [12].

For prophylactic purposes, the PK/PD parameters of antibiotics in a specific compartment of the body can be used as a surrogate for antimicrobial efficacy on the basis of the minimal inhibitory concentration (MIC) values of the expected or confirmed bacterial pathogens. However, the pharmacokinetics of drug penetration into the rigid, inflamed, and often poorly vascularized bone or joint tissues are not well characterized for all antibiotics of clinical relevance in orthopedics and trauma. There are some antibiotics that have been described to have higher bone/serum level ratios (e.g., clindamycin, the quinolones, the aminoglycosides, linezolid, doxycycline, fosfomycin, daptomycin, or rifampicin). Others, by contrast, seem to show poorer bone penetration, including e.g., the penicillins and many cephalosporins [13]. For treatment purposes, minimal biofilm eradication concentrations (MBEC) with much higher values than MIC are clinically more relevant, but often not known.

In order to overcome the limitations of systemically applied antibiotics, delivery of antibiotics directly to the site of infection has gained popularity. The advantages of a direct local administration route are obvious: in situ delivery of antibiotics increases the drug concentration in the affected compartment to levels that cannot be achieved via systemic use, limits drug-associated adverse systemic effects, and reduces widespread antimicrobial resistance development because of low circulation exposure. A strong proof of this concept has come from the recently finished multi-center open-label clinical trial SOLARIO [14]. This study demonstrated that it is possible to reduce the current therapy of four weeks or longer lasting systemic antibiotic administration in orthopedic infections to seven days or less in the presence of antibiotic-eluting biomaterials without compromising the treatment success. Not surprisingly, the non-inferior short antibiotic therapy came along with the additional benefit of fewer drug-related side effects and less antibiotic consumption [15]. This may ultimately lead to better treatment adherence, lower risk of antibiotic resistance development, and lower overall antibiotic costs.

## 2. Modalities of Local Antibiotic Delivery

In principle, the methods of local antibiotic delivery can be classified into three different categories. The simplest technique is the direct sprinkling of antibiotic powder into the open wound before wound closure. The second technique includes the implantation of a wide range of different antibiotic-loaded biomaterials, which elute the admixed drugs over time in situ. The third modality consists of direct injection/infusion of liquid antibiotics into the affected bone or joint space via the use of needles/cannulas or catheters (see Figure 1).

### 2.1. Local Vancomycin Powder Administration

With positive reports from spinal surgery suggesting a prophylactic benefit from sprinkling vancomycin powder directly onto the surgical wound or implant site [16,17], this practice has also become popular in arthroplasty. The rationale behind the topical vancomycin powder administration (typical dosage in the range of 0.5 to 2 g) is the decontamination of the exposed surgical site at the time of wound closure from gram-positive bacteria. However, controversies remain about this practice, and the evidence in arthroplasty is still weak due to the rather low quality of the predominantly observational studies in this indication. Two recently published meta-analyses in the field provide good examples of how much the benefit–risk evaluation of this practice may differ depending on parameters such as quality of the included clinical studies, definition criteria of surgical site infections, type of surgeries, and infection follow-up periods, among others. Xie et al. [18] concluded in their meta-analysis based on 24 studies with mixed designs (20 retrospective cohort studies and 4 randomized clinical trials) that topical vancomycin administration in the wound is effective in reducing both superficial and deep PJI. By contrast, the meta-analysis of Saka et al. [19], which evaluated exclusively high-quality randomized clinical studies (eight studies in total), arrived at the opposite conclusion. In the latter analysis, there was no statistical difference in the vancomycin powder group in reducing overall surgical site infections compared to the controls [19]. In addition, concerns with topical vancomycin sprinkling before wound closure exist regarding possible adverse tissue effects on osteoblast function [20] or an increase in wound healing complications [21,22]. In summary, the evidence for this practice appears to be too inconclusive to recommend this practice as routine.

### 2.2. Non-Resorbable Biomaterials as Antibiotic Carrier

#### PMMA-Bone Cement

Introduced by Buchholz et al. more than 50 years ago [23], antibiotic-loaded bone cement (ALBC) made of a mixture of different polymethylacrylate (PMMA) polymers with admixed antibiotics is commonly applied in cemented arthroplasties of the hip and knee joints. The purpose of using ALBC in primary and aseptic revision arthroplasty is prosthesis fixation and the prevention of deep surgical site infections at the same time. Antibiotic release from bone cement is based on reciprocal diffusion and is divided into two different phases [24,25]. In the first 24 h, the initial release of antibiotics, called burst release, is a quick response after implantation when the antibiotics of the surface cement layer quickly dissolute into the surrounding space after contact with the aqueous environment. This leads to a very high local concentration of antibiotics exceeding the MIC values of the common PJI bacteria by a factor of up to 100-fold and more [26]. The second phase, called sustained release, follows after 1–2 days and results in a progressively decreasing antibiotic elution over the following days and weeks. During this period, antibiotics entrapped in more inner cement layers are released when body fluids penetrate deeper into the cement by a network of microscopical cracks, voids, and pores as a result of cement compression and friction [27]. However, this pharmacokinetic release pattern of ALBC is variable and depends on many factors, including the concentration, molecular size, and solubility of the admixed drug, hydrophilicity of the cement matrix polymer formulation, mixing technique, and the anatomical spaces [28,29]. The PMMA matrix can be further improved for drug elution by increasing the water-absorbing properties, the overall porosity, and the surface area of the cement [29,30].

The advantage of bone cement as an antibiotic carrier is the dual function of the biomaterial, providing immediate mechanical fixation of the prosthesis in the joint and eluting simultaneously antimicrobial drugs for infection prophylaxis if contamination has occurred. In fact, its ability to bear immediate load is not shared by any other commercially available antibiotic carrier. This functional duality explains why bone cement is among the most frequently used and best studied biomaterials in orthopedics.

Antibiotics can also be admixed in higher doses to PMMA in order to reinforce the prophylactic efficacy or even enhance infection eradication in septic revision surgeries [29,31]. The potential benefits of ALBC use have been described in numerous registry and clinical studies [32,33,34,35]. There are rules and limitations on which substances can be admixed to PMMA. The drugs must be water-soluble, heat-stable because of the high cement polymerization temperatures, must not interfere with the chemical reaction, and, preferably, should be in powder form, broad-spectrum, and bactericidal in activity [29,36]. The aminoglycosides gentamicin or tobramycin are among those drugs that best fulfil these requirements, making them the antimicrobial ingredients of choice in most commercially available bone cements on the orthopedic market. It is further helpful to differentiate low-dose single ALBC (typically loaded with 0.5 to 1 g of an aminoglycoside per 40 g cement powder) from high-dose dual (typically containing two antibiotics in the range of 2 to 4 g or sometimes more per 40 g cement powder). While surgeons typically use low-dose single ALBC in routine primary arthroplasty procedures, they may decide to use the dual high-dose ALBC for surgeries involving a higher infection risk patient, such as a geriatric femoral fracture, or revision patients [37].

There are three commercially available versions of high-dose dual ALBC, either containing combinations of gentamicin and clindamycin, gentamicin and vancomycin, or erythromycin and colistin. In addition, specially customized high-dose antibiotic-cement mixtures can be manually manufactured in the theatre based on the expected or known pathogen profile. With the objective to provide guidance on which and how much of an antibiotic can be admixed to cement, recommendations for this admixing practice in special situations have been issued [38,39].

Dual ALBC formulations have the advantage of an often enhanced mutual drug release profile, leading to a longer-lasting and broader antimicrobial effect. This has been evidenced by several in vitro biofilm inhibition studies [40,41,42]. In vivo, the follow-up of intra-articular drug levels over time is hampered by the limited time window for which drains are left in situ (usually only for 24–72 h). Therefore, the analysis of explanted ALBC spacers, joint aspiration at spacer removal, or the determination of antibiotic levels in the spacer membrane may alternatively provide answers to the questions of how long ALBC can inhibit the growth of intra-articular bacteria in vivo. Indeed, there are several reports that revealed sufficient antibiotic concentrations above MIC for a usual spacer period of 6–12 weeks [43,44,45].

When comparing the PJI rate in patient cohorts who were allocated to receive either single ALBC or dual ALBC during hemi- or revision arthroplasty, recent studies have suggested that certain patients at high risk of infections may benefit from the dual local antibiotic prophylaxis protocol [37,46,47,48,49].

Opponents of the routine use of ALBC in arthroplasty repeatedly point to the risk of developing antibiotic resistance. In fact, it has been shown that antibiotic-persistent bacteria are able to recolonize the cement and the prosthesis surface if they manage to survive the “hit hard and hit early-phase” during the initial burst release of the antibiotics from cement [50,51]. Failure to prevent infections with, e.g., gentamicin-loaded bone cement might therefore be more often associated with an environment where gentamicin-resistant bacteria with high MIC levels are frequent. A deeper look into the MIC distribution patterns of hundreds of clinical isolates of bacteria with relevance for PJI in the European Committee on Antimicrobial Susceptibility Testing (EUCAST) source shows that coagulase-negative staphylococci are the most problematic pathogens in this respect, with gentamicin resistance rates of up to 60% based on the epidemiological breakpoint (0.25 µg/mL). However, the same EUCAST data sources demonstrate that at gentamicin concentrations which can be expected during the burst release phase, most of these so-called resistant strains are still susceptible because of the strong concentration-dependent effect of the aminoglycosides [52].

Table 1 shows a heatmap which visualizes, from the colors green for high activity to red for no activity, to what extent the reported peak elution levels of the commercial bone cement eluted antibiotics gentamicin, clindamycin, and vancomycin may inhibit different gram-positive and -negative bacteria of relevance in orthopedic infections. For this purpose, not only the MIC breakpoints for the classification of resistant and susceptible, but also the EUCAST-reported MIC distribution patterns of thousands of clinical isolates of these bacteria [52] were compared with the expected peak concentrations of the antibiotics after elution from the cement [53,54]. The plot suggests that Staphylococci, being the most dominant pathogens in bone and joint infections [55], are best covered by combining gentamicin and vancomycin (commercially available, e.g., in the bone cement COPAL G+V^®^, Heraeus-Medical, Wehrheim, Germany). In cases of suspected or confirmed presence of streptococci or anaerobic cutibacteria, a combination of gentamicin and clindamycin is also a good alternative (commercially available in the bone cement COPAL G+C^®^, Heraeus-Medical, Wehrheim, Germany).

### 2.3. Resorbable Antibiotic Carriers

Several other carriers beyond bone cement have been developed to allow the in-situ delivery of antibiotic drugs after implantation without the need for biomaterial removal. In fact, their resorbable nature makes them particularly attractive as dead space filling and antibiotic delivery material in surgeries when there is no staged infection treatment planned. These biomaterials include autologous or allogenic bone grafts, ceramic bone graft substitutes based on, for example, calcium sulfate or calcium phosphate/hydroxyapatite, bioglass, hydrogels for implant coatings, or biopolymers derived from plants and animals. All these carriers have different material formulations, degradation profiles, and can be mixed with different antibiotics. As previously described for bone cement, these biomaterials can be either mixed with antibiotics of choice in the theatre or purchased as gentamicin/tobramycin and/or vancomycin containing pre-loaded gels, pastes, suspensions, or beads. Due to their self-setting properties after contact with aqueous solutions without high curing temperatures, the range of drugs that can be admixed is broader than for bone cement. The primary function of bone grafts or bone substitutes is the filling of dead spaces and large defects with bone repair, owing to the osteoconductive/osteoinductive properties in balance with bone tissue regeneration processes.

#### 2.3.1. Antibiotic-Loaded Auto- and Allografts

Autologous bone grafts are the gold standard for bone void filling as they best meet the mechanical and biological requisites for an optimal defect-filling material in bone [56]. However, because of the need to harvest the bone grafts from a donor source within the human body, the technique of autologous bone transplantation sometimes causes substantial donor-site morbidity. An alternative option is morselized bone allograft derived from bone banks or commercially processed demineralized bone matrix [57]. Both autologous and allogenic bone chips can be impregnated with antibiotics for infection prophylactic purposes. This has also been demonstrated by a recently published study in which acetabular revision patients received morselized bone chips derived from allogenic femoral heads of the hospital tissue bank, which had been soaked in theatre with 1000 mg of vancomycin hydrochloride in a solution of 6 mL (480 mg) tobramycin solution. The in situ antibiotic concentrations measured before implantation and after impaction grafting at the damaged acetabulum were shown to remain well above MIC for all tested clinical isolates and even for highly tobramycin-resistant *Staphylococcus epidermidis* strains [58].

#### 2.3.2. Antibiotic-Loaded Calcium Sulfate

Calcium sulfate (CaSO_4_) is a simple carrier material that is soft and does not cause relevant third-body wear on osteosynthesis material or on prosthetic components. The carrier is available as cement, which can be shaped into beads of different sizes to suit a variety of clinical applications in bone and joint infections. When customized by adding antibiotics (often soaked in theatre in a solution of 1000 mg of vancomycin hydrochloride or tobramycin), it has been shown to successfully inhibit biofilm formation and eradicate even established biofilms [59,60]. The same authors have also demonstrated that staphylococcal biofilm-active concentrations of vancomycin may be obtained locally for at least 2 weeks. CaSO_4_ degradation over the following weeks after implantation allows release of up to 100% of the entrapped, making this antibiotic carrier particularly attractive in those orthopedic infections where load-bearing aspects are not relevant.

In the field of revision arthroplasty, there are a few clinical studies reporting reduced (re)infection rates with the use of CaSO_4_ (loaded with vancomycin alone or with a combination of vancomycin & tobramycin) when using them either in DAIR procedures [61,62,63] or as coating material of femoral stems in second-stage reimplantation after PJI [64].

However, the kinetics of rapid CaSO_4_ degradation within 2–3 months post-surgery result in the possible complication of delayed wound healing and aseptic wound leakage. A recent meta-analysis evaluated the prevalence of such during treatment of chronic OM and found a rate of 20% [65]. The quick dissolution of CaSO_4_ may also explain why the number of bone fractures during chronic OM treatment was higher with this biodegradable carrier compared to a slower dissolving composite biomaterial made of a mixture of 60% CaSO_4_ and 40% hydroxyapatite (HA) [66]. The authors therefore speculated that new bone formation did not match the degradation kinetics in those patients receiving CaSO_4_-based bone void fillers. Caution should also be taken when implanting high loads of CaSO_4_ beads, since cases of life-threatening hypercalcemia were sporadically reported in those settings [67]. There is currently one commercial antibiotic-preloaded CaSO_4_ filler (Osteoset T^®^, Wright Medical, Amsterdam, the Netherlands, preloaded with 4% tobramycin) on the market. Other commercial CaSO_4_-fillers (Stimulan^®^, Biocomposite Ltd., Keele, Staffordshire, UK; Synthecure^®^, Heraeus-Medical, Wehrheim, Germany) are also often used as an antibiotic carrier in orthopedics and trauma after soaking the CaSO_4_ powder with varying antibiotic solutions in theatre.

#### 2.3.3. Antibiotic-Loaded Calcium Phosphate/Hydroxyapatite (Alone or as Part of Composite Material)

Calcium phosphates (CaPs) or their naturally occurring form, hydroxyapatite (HA), are the main components of bone and teeth. Following the logic that damaged tissue can best be repaired by a substance with close resemblance, biomaterials based on CaPs were already proposed for fracture treatment over a hundred years ago [68]. CaPs have been reported to possess osteoconductive and osteoinductive characteristics. They were even shown to aid in the osteogenic differentiation of stem cells, making them a more ideal bioactive scaffold material for bone repair and regeneration [69,70]. Since the placement of plain CaP cement is contraindicated in infected bone voids because of the high risk of bacterial colonization, surgeons have started to impregnate this material with antibiotics in pathologies such as OM or FRI. There is a wide range of publications describing the patterns of drug elution and bacterial inhibition of manually antibiotic-admixed CaP or HA cement, either alone or in combination with other bioceramics. Again, the most frequently analyzed antibiotics were the aminoglycosides gentamicin or tobramycin and the glycopeptide vancomycin [71,72,73]. The commercially available ceramic cement Cerament G^®^ (BoneSupport AB, Lund, Sweden) is made of a mixture of 40% HA and 60% CaSO_4_ and preloaded with 17.5 mg gentamicin per ml paste. Cerament V is preloaded with 66 mg vancomycin. The manufacturer claims a high burst of local antibiotic elution with sustained release above MIC for at least 28 days, with no evidence of renal impairment for the overall deposited gentamicin amounts of up to 525 mg [74,75]. In a prospective series of 100 chronic OM patients, treated in a single-stage protocol with the use of such a high-delivery local antibiotic ceramic carrier, McNally et al. [76] reported very good infection-free outcome numbers of 94% during an observation period until 6 years. He further concluded from his observations that the method can be used in a wide range of patients, including those with significant comorbidities and non-unions [76].

#### 2.3.4. Bioglass

Bioactive bioglass shows measurable antimicrobial activity and appears to support bone healing in bone and joint infections. In vitro reports on S53P4 bioglass (Bonalive^®^, Bonalive Ltd., Turku, Finland) demonstrate broad-spectrum activity against multidrug-resistant and biofilm-forming pathogens through elevated pH, ion release, and osmotic effects—even in the absence of prior antibiotic loading [77]. Material studies suggest an intrinsic non-antibiotic mechanism of action on bacteria leading to the disruption of bacterial cell membranes, inhibition of bacterial growth, and prevention of biofilm formation [78]. Such effects are particularly valuable when dealing with antibiotic-resistant bacteria. 

In animal models of tibial osteomyelitis, antibiotic-loaded formulations in the form of vancomycin- and teicoplanin-loaded borate bioglass achieved complete infection eradication and induced bone regeneration after conversion to hydroxyapatite [79,80]

Human studies with chronic osteomyelitis patients and septic non-unions have reported infection control rates with the use of borate-based bioglass as an antibacterial bone void filler ranging from 78% to 92% [81]. It was further found to be safe and stimulating a positive growth response in the remaining healthy bone at the bone-glass interface [82,83].

#### 2.3.5. Hydrogel

Recently, a new hydrogel-based intraoperative coating method has been developed that works as a physical barrier to bacterial adhesion and that can be intra-operatively loaded with various antibacterial agents [84]. The DAC^®^ (Defensive Antibacterial Coating, Novagenit Srl, Mezzolombardo, Italy) substance is a commercial fast-resorbable hydrogel composed of covalently linked hyaluron and poly-D,L-lactide. It is supplied in a prefilled syringe containing a powder to be reconstituted at the time of surgery with a solution composed of 5 mL sterile water and antibiotic(s) of choice ranging from 20 mg/mL to 50 mg/mL. The principle has been subsequently tested in various orthopedic conditions, including primary and revision surgery, osteosynthesis in fracture repair, and implantation of megaprosthesis in oncologic surgeries. In a multicenter randomized study including 380 primary or revision patients, Romanó et al. reported a significant decrease in surgical site infection rates when comparing those patients who received DAC-coated cementless or hybrid implants vs. those without pretreatment of the prosthesis (SSI rate of 0.6% vs. 6%) [85]. Zoccali et al. [86] compared the post-surgical infection rate in a three-center case-control study involving 43 patients who received joint mega-prostheses because of bone tumors. They found no case of infections in the DAC-treated group during a follow-up of 2 years vs. 6 cases in the control group [86]. Finally, a much lower post-surgical infection rate was also observed in the DAC-treated group after osteosynthesis in closed fractures compared to the untreated controls (0% vs. 4.8%) [87].

In all clinical tests so far, the DAC coating principle has proven to be safe without differences in wound healing, clinical scores, laboratory tests, and radiographic findings between the treatment and control groups.

### 2.4. Intraosseous/Intra-Articular Injection or Infusion of Antibiotics

An innovative technology to overcome the limitations and potential drawbacks of biomaterial-bound antibiotic delivery is the direct intraosseous (IO) or intra-articular injection or infusion (IA) of antibiotics via catheters, needles, trocar cannulas, or novel irrigation/instillation devices. This delivery route offers unprecedented possibilities of antibiotic administration at permanently high levels in situ and has yielded promising first results, as shown below. It makes the surgeon more independent from the above-mentioned suboptimal pharmacokinetic features of antibiotic biomaterial elution and from possible interactions with carrier components, limiting the choice of admixed drugs. Because of the permanent flow of drugs into the affected tissues and spaces, this antibiotic delivery modality can potentially ensure antibiotic concentrations far above MIC for the entire application period for prophylactic and therapeutic purposes [88,89,90,91,92,93]. The risk of carrier material colonization by persistent bacteria is also no big concern [93]. A further advantage of this local antibiotic delivery route is the greater flexibility to immediately adjust the antibiotherapy to the profile of the pathogens once the microbiological results are available [93,94]. This makes it easier to precisely select and dose the active drug during the administration period. In summary, this technique has the potential to switch from an often more empirical local antibiotic treatment with biomaterial-based delivery to a targeted approach with positive implications for the treatment duration and antimicrobial efficacy.

It has been repeatedly shown that the IO administration of antibiotics is relatively easy, rapid to perform, and associated with very high tissue concentrations. For practical reasons, this technique is more widely used for infection prophylactic purposes in the field of total knee arthroplasty (TKA) with a single-shot administration of prophylactic antibiotics via a cannula inserted into the proximal medial tibia between the tibial tubercle and joint line [88,95]. It has been shown that IO administration provides consistently much higher tissue concentrations compared to intravenously (IV) administered prophylactic antibiotics. Young et al. reported more than 10 times higher tissue concentration of cefazolin in the patient group receiving 1000 mg of the drug upon IO injection compared to the standard group receiving 1000 mg by IV administration [96]. In a similar way, several studies comparing periarticular tissue concentrations of low-dose vancomycin (500 mg) have also shown 6–10 times higher concentrations with IO injection compared to vancomycin IV administration [89,97]. This high level of vancomycin is also maintained in revision TKA patients [98]. The efficacy of IO administration for PJI prevention in TKA patients was recently analyzed by Yu et al. [90] in a meta-analysis comprising 7 prospective randomized controlled trials and 5 retrospective cohort studies with 4091 patients. It was concluded that IO 500 mg vancomycin was more effective in reducing the occurrence of PJI compared to IV 1000 mg vancomycin (OR: 0.19, *p* < 0.001) [90]. With views on safety aspects, it was further demonstrated that the rate of vancomycin-related complications, including cases of Red Man syndrome, neutropenia, or acute kidney injury, was not different between the groups [90]. Taken together, the existing clinical evidence so far suggests that the technique of IO administration of low-dose vancomycin holds the promise to be an attractive alternative or adjunct to the trusted IV antibiotic prophylaxis in primary and revision arthroplasty. The option might be particularly interesting for those patients who are at risk of MRSA colonization or who receive cementless prostheses. Furthermore, this strategy eliminates the logistical challenges associated with timely vancomycin administration.

In contrast to the prophylactic use of IO antibiotic injection, the use of this technique in the therapeutic framework of treating bone and joint infections is still more experimental. The IA treatment approach involves, in the majority of reported cases, a Hickmann or three-branch catheter instead of a cannula, which is placed in the infected area. Its rationale is to enhance treatment efficacy by improving antibiotic penetration into the biofilm. Ji et al. presented their experiences with IA antibiotic infusion in single-stage revisions for treating culture-negative (CN) or polymicrobial PJI cases with difficult-to-treat pathogens in several reports [99,100,101]. In particular, the results of the study with chronic PJI patients suffering from refractory infections after multiple failed surgeries deserve special attention. Using a protocol with supplementary IA infusion of antibiotics (protocol of incubation of vancomycin, meropenem, and/or voriconazole in the joint for 24 h and subsequent extraction before next infusion for a mean of 16 days), they observed an infection-free 7-year survival rate of 87.6% [101]. With prior surgeries considered to be a strong predictor of treatment failure, this is an unusually high success rate for such patients in a one-stage revision procedure. In a similar way, Whiteside et al. used this technique in refractory PJI cases and in infections with the presence of methicillin-resistant pathogens and reported a high level of infection control [94,102]. Table 2 presents a summary of clinical studies and case series reporting on the experiences/clinical outcomes with the IO and IA antibiotic administration route, with further differentiation between prophylactic and therapeutic use in bone and joint infections.

This direct drug delivery route also offers the unique possibility of a “biological” infection treatment with specific bacteriophages to overcome the limitations of classic antibiotics in refractory chronic bone and joint infections with multi-drug resistant pathogens. Although still in the very experimental stage, published case reports have shown very positive clinical outcomes. The majority of patients experienced clinical resolution of repeated chronic reinfections or substantial improvement, often with documented healing of fistulas and absence of local signs of infection at follow-up [114].

## 3. Conclusions

There is good evidence for the successful use of local antibiotics in preventing and treating bone and joint infections. In present clinical practice, the in situ delivery of antimicrobial drugs is mainly based on the topical sprinkling of antibiotic powder into the wound or on biomaterial-bound antibiotic elution in situ after implantation in bone or joint cavities. However, because of the inherent pharmacokinetic limitations of these drug delivery methods, the technologies of direct intraosseous injection or intra-articular infusion have started to be explored. They hold the promise of an even more effective infection prophylaxis and treatment by targeting resistant pathogens in patients at high infection risk with continuously high drug levels. By offering a comprehensive evaluation of the current state and existing evidence base of the local antibiotic delivery techniques, this review aims to inform and guide clinical practitioners. It may also help to stimulate further research into this important area for bone and joint infection management.

## Figures and Tables

**Figure 1 microorganisms-13-02276-f001:**
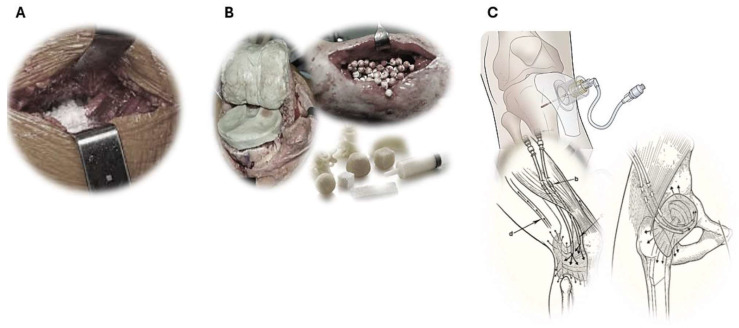
Different means of antibiotic delivery in situ. (**A**) Sprinkling of topical vancomycin powder onto the prosthesis or soft tissue before wound closure, (**B**) implantation of antibiotic-loaded biomaterials in bone cavities and joint spaces, (**C**) direct intraosseous or intra-articular administration of liquid antibiotics into bone tissues and joint spaces via injection /infusion through e.g., needles, cannulas, or catheters.

**Table 1 microorganisms-13-02276-t001:** Probabilities of prophylactic antimicrobial efficacy of the antibiotics gentamicin, clindamycin, and vancomycin eluted from commercially available pre-loaded ALBC against the most prevalent bacterial pathogens in PJI and FRI based on the EUCAST published MIC distributions.

Pathogen	Gentamicin or Tobramycin	Clindamycin	Vancomycin
Gram-positive aerobic bacteria			
Coagulase-positive Staphylococci *S. aureus* (MSSA)	+++	+++	+++
*S. aureus* (MRSA)	++	++	+++
Coagulase-negative Staphylococci e.g., *S. epidermidis*	+	+	+++
Streptococci	(+)	++	+++
Enterococci	(+)	−	++
Gram-positive anaerobic bacteria			
*C. difficile*	−	−	++
*C. acnes*	(+)	++	++
Gram-negative aerobic bacteria			
*E. coli*	++	−	−
*K. pneumoniae*	++	−	−
*P. aeruginosa*	++	−	−
*Enterobacter, Serratia*	++	−	−

Data are based on the European Annual Antimicrobial Susceptibility Testing (EUCAST) of bacterial samples and strains sent in from laboratories of different European countries [52]. Antibiotic susceptibilities of the different PJI-relevant bacteria in this graph also consider the expected higher antibiotic concentrations than MIC after peak elution from bone cement [53,54]. Local differences in pathogen susceptibilities occur and should be regularly monitored. (+++ = very high probability of a strong antimicrobial effect, ++ = high probability of a strong antimicrobial effect, + = moderate probability of a strong antimicrobial effect (+) = low probability of a strong antimicrobial effect, − = no expected antimicrobial effect).

**Table 2 microorganisms-13-02276-t002:** Overview of published clinical studies and case series using either the technique of regional intraosseous (IO) or intra-articular (IA) injection/infusion of antibiotics into bone tissue or joint space. The studies are grouped according to the primary intended use, which is prophylactic or therapeutic use. Further information on the number of included patients in each study, on the choice of the antimicrobial drug, its concentration and duration of administration, on the mode of drug delivery, and on major clinical observations/outcomes is provided in detail. The vast majority of studies have been performed in arthroplasty and prosthetic joint infections.

Authors and Study	Number of Patients	Indication and Intended Purpose	Type, Concentration and Duration of Antibiotic Administration	Route of Drug Delivery	Key Findings of Study and Clinical Outcome
Prophylactic & therapeutic use with endpoints safety & pharmacokinetics					
Spangehl et al., 2022 [103]	Randomized study with 24 patients	Prophylactic use in primary TKA Patients randomized to receive either vancomycin IV or vancomycin IO	Vancomycin IV (weight-based (15 mg/kg) vs. **vancomycin IO** (500 mg in 100 mL solution), single-shot	Intraosseous bolus injection via cannula into proximal tibia	Median vancomycin concentrations in tissue were significantly higher (5–15 times) at all time points in the vancomycin IO group
Young et al., 2013 [96]	Randomized study with 22 patients	Prophylactic use in primary TKA Patients randomized to receive either cefazolin IV or cefazolin IO	Cefazolin (1 g) IV 10 min before tourniquet inflation. Vs. cefazolin IO (1 g in 200 mL of normal saline), single shot	Intraosseous bolus injection via a tibial cannula after tourniquet inflation	Mean tissue concentration of cefazolin in subcutaneous fat was 186 μg/g in the IO group and 11 μg/g in the IV group. The mean tissue concentration in bone was 130 μg/g in the IO group and 11 μg/g in the IV group.
Young et al., 2014 [89]	Randomized study with 30 patients	Prophylactic use in primary TKA Patients randomized to receive either vancomycin IV or vancomycin IO	Vancomycin IV (fixed dose, 1 g) vs. **vancomycin IO** (250 or 500 mg), single-shot	Intraosseous bolus injection via cannula into proximal tibia	Mean tissue concentration of vancomycin in subcutaneous fat was 14 μg/g in the 250 mg IO group, 44 μg/g in the 500 mg IO group, and 3.2 μg/g in the IV group. Mean concentrations in bone were 16 μg/g in the 250 mg IO group, 38 μg/g in the 500 mg IO group, and 4.0 μg/g in the IV group.
Harper et al., 2020 [104]	Retrospective review of 119 TKA patients (100 primary and 19 revision cases)	Prophylactic use in primary and revision TKA	Vancomycin IV vs. **vancomycin IO** (500 mg in 200 mL saline solution)	Intraosseous injection of 100 mL vancomycin solution via cannula in tibial tubercle region and 100 mL in distal femur	No significant differences in the complication rate or creatinine values were identified between IO and IV groups.
Young et al., 2018 [97]	Randomized study with 20 patients	Prophylactic use in revision TKA Patients randomized to receive either vancomycin IV or vancomycin IO	Vancomycin IV (fixed dose, 1 g) vs. **vancomycin IO** (500 mg), single-shot	Intraosseous bolus injection via cannula into proximal tibia	The mean tissue concentration of vancomycin in fat samples was 3.7 μg/g in the IV group vs. 49.3 μg/g in the IO group; mean tissue concentrations in femoral bone were 6.4 μg/g in the IV group vs. 77.1 μg/g in the IO group. Vancomycin concentrations in the final subcutaneous fat sample taken before closure were 5.3 times higher in the IO group vs. the IV group.
Klasan et al., 2021 [105]	Retrospective study of 331 cases receiving IO vancomycin	Prophylactic use in TKA	**Vancomycin IO** (500 mg), single-shot in addition to weight-based cefazolin	Intraosseous injection via cannula into proximal tibia	IO vancomycin in addition to standard IV cefazolin prophylaxis in TKA is safe without significant adverse effects of vancomycin, such as acute kidney injury, red man syndrome, or neutropenia. The 90-day PJI rate was 0%, and the 1-year PJI rate was 0.2%.
Springer et al., 2024 [91]	Randomized multi-center (17 hospitals) study with 76 patients	Treatment use in chronic hip & knee PJI. Patients randomized to 2-stage exchange arthroplasty with either cyclic IA irrigation or with standard ALBC spacer in addition to IV antibiotics	Start with **tobramycin IA** irrigation using 80 mg in 50 mL of saline (1.600 μg/mL) daily with a 2 h soak followed by 30 min of vacuum to actively drain the intra-articular joint space. The patient then received hourly irrigation using 125 mg of **vancomycin IA** in 50 mL of saline with a 30 min soak and a 30 min vacuum. Patients received 22 total irrigation cycles per day (approximately 2.750 mg vancomycin/day). Duration: 7 days	Cyclic IA irrigation with instillation and evacuation of antibiotics through a short-term implantable porous titanium spacer	Both detectable vancomycin and tobramycin concentrations were well below established systemic toxicity concentrations, and no case of systemic side effects was observed. Advantage of this therapy modality is that it includes a pump that does not require manual injection of the antibiotics, and the antibiotics are removed through a vacuum system after a soak period, eliminating the concerns for fluid accumulation in the joint.
Prophylactic & therapeutic use with endpoints infection control/reinfection					
Parkinson et al., 2021 [106]	Retrospective multi-center study of 1909 cases	Prophylactic use in primary TKA	**Cefazolin IO** (1 g) in 324 patients or **vancomycin IO** (500 mg) in 391 patients, single shot, with or without supplementary IV prophylaxis	Intraosseous injection via cannula into proximal tibia	IO regional antibiotic delivery was associated with a lower risk of infection within 12 months (0.1%) compared with the risk after traditional IV administration (1.4%, relative risk = 0.10; *p* = 0.03).
Yu et al., 2024 [90]	Meta-analysis of 12 studies (7 prospective, 5 retrospective) with 4091 cases	Prophylactic use in primary TKA	Vancomycin IV (weight-based (15 mg/kg) vs. **Vancomycin IO** (500 mg), single-shot	Intraosseous injection via cannula into proximal tibia	IO vancomycin significantly increased the drug concentration in the periarticular adipose and bone tissue compared to IV vancomycin. Regarding the incidence of postoperative PJI after primary TKA, IO vancomycin was more effective in reducing the occurrence of PJI compared to IV vancomycin (OR: 0.19; 95% CI: 0.06–0.59; *p* < 0.001). No significant differences were found between the two groups in terms of postoperative pulmonary embolism and vancomycin-related complications.
Park et al., 2025 [107]	Retrospective review of 1923 cases	Prophylactic use in primary TKA	Vancomycin IV (weight-based (15 mg/kg) vs. **vancomycin IO** (500 mg), single-shot	Intraosseous injection via cannula into proximal tibia	IO group had significantly lower incidence of PJI compared to the IV group at 90 days (0.5 vs 1.6%, *p* = 0.018), 1-year (0.7 vs. 1.8%, *p* = 0.048), and 2-year (0.9 vs. 2.4%, *p* = 0.032) follow-up. In addition, there was a lower incidence of nonoperative wound complications requiring oral antibiotics in the IO group, as well as a lower incidence of acute kidney injury
McNamara et al., 2025 [108]	Retrospective review of 719 cases	Prophylactic use in aseptic revision TKA	Vancomycin IV (weight-based (15 mg/kg) vs. **vancomycin IO** (500 mg), single-shot	Intraosseous injection via cannula into proximal tibia	IO cohort with significantly lower PJI incidence compared to the IV cohort at 30 days (0.3 vs. 2.1%, *p* = 0.03), 90-day (0.9 vs. 3.1%, *p* = 0.04), and 1-year follow-up (1.6 vs. 4.9%, *p* = 0.04). There were no reported adverse reactions to vancomycin and no differences in the incidence of acute kidney injury, deep venous thrombosis or pulmonary embolism between the groups.
Christopher et al., 2024 [109]	Observational study of 117 cases	Prophylactic use in aseptic revision TKA	**Vancomycin IO** (500 mg) in conjunction with IV cephalosporins or clindamycin.	Intraosseous injection via cannula into proximal tibia	The rate of PJI was 0% at 3 months postop. Follow-up at 1 year was obtained for 113 of the 117 revision TKAs, and the PJI rate remained 0%. The rate of PJI at the final follow-up of ≥1 year was 0.88%.
Ji et al., 2019 [99]	Observational study of 126 cases	Treatment use in hip PJI Single-stage w/o prior patient selection	**Vancomycin IA** for MDR gram-pos. bacteria **Vancomycin + Imipenem IA** for polymicrobial organisms with gram-neg. bacteria **Fluconazole/Voriconazole IA** for fungi Conc. Vancomycin: 500 mg/days Conc. Imipenem: 500 mg/days Conc. Fluconazole/Voriconazole: 100–200 mg/days Mean duration:16–18 days	Catheter-based intra-articular infusion	Total infection-free cases were 89.2% at a mean follow-up time of 58 months. The success rate in patients with multidrug-resistant organisms was 84.2%.
Ji et al., 2020 [100]	Observational study of 51 cases	Treatment use in hip and knee culture-negative PJI Single-stage	500 mg **Vancomycin IA** + 500 mg **Imipinem IA** per day, alternately in the morning and afternoon	Catheter-based intra-articular infusion	No additional medical treatment for recurrent infection for 90.2% of cases at a mean of 53.2 months was needed. Impaired kidney function observed in 2 patients.
Ji et al., 2022 [101]	Observational study of 78 cases	Treatment use in hip and knee PJI in patients with multiple prior surgical interventions because of infection recurrence	500 mg **Vancomycin IA,** 500 mg **Imipenem IA**, or 100 mg **voriconazole IA** per day. The antibiotic solution was soaked into the joint for 24 h for a mean of 16 days	Catheter-based intra-articular infusion	The seven-year infection-free survival was 87.6% for all patients. No significant difference in infection-free survival was observed between hip and knee PJIs
Li et al., 2023 [110]	Observational study of 32 cases	Treatment use in hip & knee PJI in patients with gram-negative pathogens	500 mg **Imipenem IA** for single gram-negative PJI per day 500 mg **Vancomycin** + 500 mg **Imipenem IA** for polymicrobial PJI with gram-neg	Catheter-based intra-articular infusion	Of 32 cases, treatment failed to eradicate infection in only three cases (9.4%), at a mean follow-up of 55.1 months.
Bruyninckx et al., 2024 [92]	Meta-analysis of 15 articles, encompassing 631 PJIs in 626 patients, all retrospective studies or case series	Treatment use in hip & knee PJI. 79.1% of cases were treated in single-stage revisions with adjuvant IA antibiotic infusion, 12.2% in single-stage revisions with stand-alone IA infusion, 5.7% in DAIR and 3.0% in two-stage revisions	In vast majority of cases **vancomycin or gentamicin IA with varying protocols re dosage and duration of administration**	Catheter-based intra-articular infusion	Mean duration of IA antibiotic infusion was 19 days (range 3–50). An overall failure rate of approximately 11% was found. In total 117 complications occurred, 71 were non-catheter-related and 46 were catheter-related. The most common catheter-related complications were premature loss of the catheter and elevated blood urea nitrogen and creatinine levels. 17 of the 18 patients had control of infection and achieved durable fixation and a closed wound. 1 case needed re-treatment, but remained then asymptomatic for 28 months post-op.
Whiteside et al., 2012 [102]	Observational study of 18 patients	Treatment use in knee PJI after failed one- and two-stage revision	**Vancomycin or gentamicin IA**. Starting dose of 100 mg vancomcin or 20 mg gentamicin in 3 mL. The concentration and volume were increased daily if the wound remained sealed and quiescent. Dosage was increased to 500 mg vancomycin or 80 mg gentamicin in 8 mL saline. The dose was given every 12 or 24 h for 6 weeks. The injection was alternated between the two catheters to keep them open. The catheters were not flushed.	Hickman double catheter-based infusion into the intraarticular space	
Whiteside et al., 2011 [94]	Observational study of 18 patients	Treatment use in knee PJI with MRSA pathogens	**Vancomycin IA.** 500 mg vancomycin in 10 mL saline solution once or twice daily for 6 weeks; no administration of IV antibiotics after the first 24 h.	Hickman double catheter-based infusion into the intra-articular space	Infection was controlled at last follow-up (42 months) in all but 1 patient with a recurrence of the MRSA pathogens. Re-infection in this case appeared controlled after second intervention until end of follow-up period.
Whiteside and Roy, 2017 [111]	Observational study of 30 patients	Treatment use in hip PJI (21 with chronic PJI treated with single-stage exchange, 9 with late acute PJI, treated with DAIR)	**Vancomycin, gentamicin IA** (in one case). Infusion of drugs was started as soon as the incision was sealed and dry. The dose was increased gradually. Beginning dose was 100 mg for vancomycin in 3 mL water. If tolerated (no wound drainage), daily increase to maintenance dose of 400–500 mg in 5 or 6 mL water. The starting dose for gentamicin was 10 mg in 3 mL saline, and the maintenance dose was 40 mg in 4 mL normal saline. Duration of antibiotic infusion: 6 weeks	Hickman double catheter-based infusion into the intra-articular space	95% infections in patients with single-stage revision for chronic PJI remained free of infection at a mean follow-up of 63 months- One case grew *Candida albicans* in the operative cultures and remained free of signs of infection after re-revision followed by infusion of fluconazole. The nine acute PJI cases treated with DAIR and head/liner change all remained free of signs of infection at a mean follow-up of 74 months. No patient had evidence of permanent renal damage. None developed a chronic fistula or had significant drainage from the catheter site.
Hieda et al., 2025 [112]	Observational study of 32 cases	Treatment use in hip PJI (all patients treated with DAIR, including 11 chronic PJI cases. In total, 22 patients were treated with DAIR & CLAP, 10 patients with DAIR only.	**Gentamicin (amikacine or arbekacin) IA** were prepared in saline at 1.2 (or 2 mg/mL, respectively). These solutions were continuously administered at a rate of 2.0 mL/h for 24 h daily using a continuous precision pump through a Salem Sump tube. In two fungal cases **micafungin IA** was diluted in saline to 50 μg/mL and administered at the same rate and duration. Mean duration of local antibiotic infusion: 15.5 days	Continuous local antibiotic perfusion (CLAP) with a double-lumen tube (antimicrobial administration on one side and negative pressure application on the other). It features multiple holes for suction under continuous negative pressure to prevent blockage.	The implant survival rate after DAIR surgery supplemented with CLAP was 90.9% (20 of 22 cases), and without CLAP was 70.0% (7 of 10 cases). In two cases in the CLAP group, the implants were removed and replaced because of recurrent peri-implant infection.
Kosugi et al., 2022 [113]	Restrospective study of 9 cases	Treatment use in fracture-related infections of the lower limb	**Gentamicin IO** (60 mg in 50 mL NaCl = 1200 μg/mL) 2 mL/h infusion speed Mean duration:17 days	Continuous local antibiotic perfusion (CLAP)	Observational study in refractory FRI with difficult-to-treat pathogens. Implants were preserved until bone union was achieved. Infection was suppressed in all cases (in some cases by repeating this method). No side effects were observed.
Lawing et al., 2015 [93]	Restrospective study of 351 cases	Prophylaxis of fracture-related infections (all open fractures)	**Gentamicin or Tobramycin IO** (intervention group) in addition to IV antibiotics. Control group with only IV antibiotics, Injection of 80 mg of the aminoglycoside, diluted in 40 mL of normal saline (2 mg/mL) by inserting the needle down to the bone and implant after wound closure. In some patients with type-II and III fractures, an additional catheter was placed within the wound and irrigations with a 0.5-mg/mL mixture of aminoglycoside and normal saline were performed every six hours. Duration. 3–5 days	Direct injection into infected dead space (plus catheter-based delivery in some cases of Gustilo type II or III fractures)	The deep and superficial infection rate in the control group was 19.7% (36 of 183 fractures), and 9.5% (16 of 168 fractures) in the intervention group (*p* = 0.010). When comparing only the deep infections, the infection rate in the control group was 14.2% vs. 6.0% (*p* = 0.011). After multivariate analysis to adjust for possible confounding factors, the administration of local antibiotics was found to be an independent predictor of lower infection rates in both deep and superficial infections (odds ratio, 2.6) and deep infections only (odds ratio, 3.0). The use of local antibiotics did not have an impact on nonunion.

## Data Availability

No new data were created or analyzed in this study. Data Sharing is not applicable.

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
