# Peer review of "Current Concepts of Local Antibiotic Delivery in Bone and Joint Infections—A Narrative Review of Techniques and Clinical Experiences"

_microorganisms, 2025, doi:10.3390/microorganisms13102276_

Round 1

Reviewer 1 Report

Comments and Suggestions for Authors

Dear Authors,

Thank You for a pleasure to read Your work.

I have several offers to improve Your manuscript.

Sincerely, Reviewer

Title

Please, add ‘implant-related’ as You feel about this infection type in the abstract.

All acronymous must be written in the full form for first time use. Please, write the aim of Your article.

Keywords: please, check them according to MeSH.

Introduction

It is not relevant to tell about hypothesis exception/rejection for literature review as it is not clinical study.

Also, it is better to replace the aim in the end of section.

Subsection 2.

Please, add links for lines 84-96

Line 111: please, write what PJI is.

Subsection 2.3 Please, add the links for them.

Subsections 2.3, 2.4 Please, consider it as separate section due to the difference in meaning with other subsections.

Lines 363-380 require their links.

Table 2: please, make the subsections with title of study type and reorganize all studies according to their date.

Author Response

I greatly appreciate the very constructive comments of the reviewer. Indeed, the implementation of the suggestions have led to a noteable improvement of the article.

Title & Abstract

Comment 1: Please, add ‘implant-related’ as you feel about this infection type in the abstract.

Answer: Done. „Implant-associated“ has been changed in the abstract of the revised manuscript into „Implant-related bone and joint infections…“ (line 9)

Comment 2: All acronyms must be written in the full form for first time use. Please, write the aim of your article.

Answer: Done in the abstract of the revised manuscript…„“Antibiotic-loaded biomaterials, such as Polymethyl Methacrylate (PMMA) cement…“. (line 12)

Good point to add the aim of the article in the abstract: Done „This articles summarizes the knowledge of when, how and in which clinical settings the different modes and philosophies of local antibiotic delivery work best with the aim ot provide surgeons and infectious disease physicians guidance in clinical practice. (lines 21-24)

Comment 3: Keywords: please, check them according to MeSH.

Answer: Done and replaced according to the MeSH descriptor data 2025.

New Keywords in the Manuscript: Prosthesis-Related Infections, Infectious Bone Diseases, Local Anti-Infective Agents, Prophylaxis, Treatment, Biocompatible Materials, Bone Cements, Bone Substitutes, Calcium Sulfate, Calcium Phosphate, Bioglass, Hydrogels, Intra-Articular, Intraosseous (lines 25-28)

Introduction

Comment 4: It is not relevant to tell about hypothesis exception/rejection for literature review as it is not clinical study. Also, it is better to replace the aim in the end of section.

Answer: With the greatest respect, I disagree here partly with the comment. In the introduction I have not put forward a hypothesis of my own, but summarized the current state of our knowledge pointing to the limitations of what is actually done and what can be improved. However, I have changed the word hypothesis to assumption in the following sentence: „…There is scientific support for the assumption that antibiotics given at higher dose are related to better infection control…(lines 53-54)

The last sentence…“This may ultimately lead to better treatment adherence, lower risk of antibiotic resistance development and lower overall antibiotic costs.“  does not describe my own interpretation and aim of the clinical study review, but refers to the previously described findings of the SOLARIO study which may have a great impact on future treatment strategies.

Subsection 2.

Comment 5: Please, add links for lines 84-96.

Answer: I am not sure whether links are already needed here. The lines 84-96 describe in very general terms which modalities of local antibiotic delivery exist with the corresponding figure and legend below. In the following subsections 2.1–2.4 the individual methods are then described in detail providing information on the type of delivery material, type and dosage of antibiotic added and the corresponding clinical experiences with each one of them. Very important, the literature links to all these methods and clinical informations are provided then in the subsections 2.1-2.4 for further in-depth information.

Comment 6: Line 111 - please, write what PJI is.

Answer: The term PJI for Prosthetic Joint Infection has already been introduced in Line 35 of the Introduction.

Comment 7: Subsection 2.3 - Please, add the links for them.

Answer: see also comment 5. Again, the lines 227-243 are meant as an introduction into this sub-chapter of resorbable biomaterial carriers describing in very general terms which options are possible. In the following subsections 2.3.1 to 2.3.5 the resorbable carrier categories are introduced one by one with their corresponding clinical experiences and, of course, corresponding literature citations.

Comment 8: Lines 363-380 require their links.

Answer: Done, I agree that citation links are here required, even if the most important studies in this category are later listed separately with their corresponding citations. The corresponding links have been inserted into the text:

„... Because of the permanent flow of drugs into the affected tissues and spaces thisantibiotic delivery modality can potentially ensure antibiotic concentrations far above MIC for the entire application period for prophylactic and therapeutic purposes [89,91,94,104,110,114]. The risk of carrier material colonisation by persistant bacteria is also no big concern [114]. A further advantage of this local antibiotic delivery route is the greater flexibility to immediately adjust the antibiotherapy to the profile of the pathogens once the microbiological results are available [98,114]. This makes it easier to precisely select and dose the active drug during the administration period.“

Comment 9 - Table 2: please, make the subsections with title of study type and reorganize all studies according to their date.

Answer: Great suggestion for better visibility. The table has been rearranged according to the date of each listed study and grouped into subsection 2.4.1 „Prophylactic & therapeutic use with endpoints safety &  pharmacokinetics“ and subsection 2.4.2 „Prophylactic & therapeutic use with endpoints infection control/reinfection“ (see revised table 2 in supplementary material to the revised manuscript)

Reviewer 2 Report

Comments and Suggestions for Authors

This is an interesting manuscript detailing current methods of antibiotic delivery for treatment of orthopaedic infections. The work cited centers around clinical/pre-clinical work, and is appropriate for the breadth of this review. I found it interesting and was looking up references relative to some of the procedures that were described. However, as part of the review, it should be mentioned that MIC is an in vitro measurement and it is not certain if above MIC is an adequate benchmark for clinical efficacy. Further, it should even be discussed that how much antibiotic and how long treatment should occur still remains unknown, especially when differences in bacterial phenotype (small colony variants, cheaters, persisters) are considered. Overall, I did find new studies and found it a good high level review. 

Author Response

Comment: This is an interesting manuscript detailing current methods of antibiotic delivery for treatment of orthopaedic infections. The work cited centers around clinical/pre-clinical work, and is appropriate for the breadth of this review. I found it interesting and was looking up references relative to some of the procedures that were described. However, as part of the review, it should be mentioned that MIC is an in vitro measurement and it is not certain if above MIC is an adequate benchmark for clinical efficacy. Further, it should even be discussed that how much antibiotic and how long treatment should occur still remains unknown, especially when differences in bacterial phenotype (small colony variants, cheaters, persisters) are considered. Overall, I did find new studies and found it a good high level review

Answer: Thank you very much for taking the time to review this manuscript. Your comment is very encouraging and highly appreciated. I do strongly agree with you that the MIC values are derived from in vitro tests and it is not certain whether they are adequate benchmarks for clinical efficacy in all aspects. This is certainly a lot more true if we look at clinically meaningful antimicrobial effects in the context of treatment of biofilm-associated bone and joint infections because of the particularly resistant phenotype of the biofilm-embedded and sometimes intracellular bacteria. Indeed, these small colony variants, cheaters or persister bacteria are the reason why simple antibiotic treatment of those infections often does not result in infection eradication. MBEC (Minimal Biofilm Eradication Concentration) is here certainly the more adequate term, although biofilm-effective concentrations are often not defined. Therefore, successful treatment of biofilm-associated bone and joint infections always requires a combination of surgical, mechanical and chemical steps with antibiotics enhancing the probabilities of infection control or infection eradication.The higher the in situ antibiotic concentrations, the higher the probability of treatment success. This is the reason why those local antibiotic delivery philosophies are so popular among orthopaedic and trauma surgeons.

In general, the focus of this review article lies more on the prophylactic effect of the antibiotics eluted from the local carrier material. This is particularly true for the use of antibiotic-loaded bone cement which purpose is the „decontamination“ of the surgical site and the prevention of (re)colonization of the cement and implant. I think in the prophylactic context MIC values as surrogates for the assessment of a possible antimicrobial effect are here still justified.

In short, in order to make your good points clearer in the manuscript, I have made changes to the relevant passages in the text:

Introduction

„…Cure of the most serious bone and joint infection pathologies, such as osteomyelitis (OM), fracture-related infections (FRI) or prosthetic joint infections (PJI), is often hampered by the presence of antibiotic-resistant bacteria encased in the protective extracellular matrix of biofilms around implants and at the site of necrotic bone tissue [2,3]. Indeed, the chronic nature of biofilm infection with antibiotic refractory persister and intracellular bacteria makes the treatment of such infections very demanding and, ideally, requires a multidisciplinary approach...(lines 33-40)

For prophylactic purpose, the PK/PD parameters of antibiotics in a specific compartment of the body can be used as a surrogate for antimicrobial efficacy on the basis of the minimial inhibitory concentration (MIC) values of the expected or confirmed bacterial pathogens...For treatment purpose, minimal biofilm eradication concentrations (MBEC) with much higher values than MIC are clinically more relevant, but often not known..“ (lines 60-71)

Main text

Table 1. Probabilities of a prophylactic antimicrobial efficacy of the antibiotics gentamicin, clindamycin and vancomycin eluted from commercially available pre-loaded ALBC against the most prevalent bacterial pathogens in PJI and FRI on basis of the EUCAST published MIC distributions. (line 218)
